# Do BitNet Gains Survive Synthesis?
# An Implementation-Aware Benchmark of
# Low-Precision FPGA Inference for Jet Classification

**Dorian Sloot**[1,2]

[1]*Atominstitut, TU Wien*
[2]*Marietta Blau Institute for Particle Physics (MBI), Austrian Academy of Sciences*
*Vienna, Austria*

## Abstract

Fast machine learning requires balancing predictive performance against strict latency and hardware constraints, particularly in real-time scientific inference where throughput and low latency must be maintained under limited FPGA resources. This study presents a controlled benchmark of low-precision multilayer perceptrons for FPGA-based jet classification using the public OpenML `hls4ml_lhc_jets_hlf` dataset. Binary and ternary BitNet-style networks are compared against dense, QKeras, HGQ, and XGBoost baselines under a unified HLS C-synthesis workflow targeting a Xilinx VU13P FPGA at a 5 ns clock period and initiation interval II=1. The benchmark evaluates predictive performance, synthesized latency, and FPGA resource usage for the binary task of light-flavour QCD-like jets versus jets labelled as $W$, $Z$, or top. The results show that BitNet-style scaling improves predictive performance relative to naive binary and ternary QKeras baselines, but does not universally yield the best FPGA implementation. HGQ provides the strongest neural resource-efficiency tradeoff, while unrolled BDTs achieve the lowest synthesized latency, demonstrating that implementation details can dominate theoretical low-bit operation-count reductions.

## 1 Introduction

Fast machine learning (FastML) is central to real-time scientific inference, where predictive performance must be balanced against strict latency, throughput, and hardware-resource constraints. In high-energy physics (HEP), trigger systems provide a representative example of this challenge, motivating aggressive low-precision inference.

Quantization-aware training (QAT) is a standard strategy for reducing arithmetic complexity and memory footprint in FPGA-deployed neural networks. Frameworks such as QKeras [1] and High Granularity Quantization (HGQ) [2] provide efficient fixed-point implementations with strong predictive performance. BitNet extends this paradigm by constraining weights to binary or ternary values while preserving dynamic range through learned scaling factors [3–5].

Most low-bit model comparisons emphasize arithmetic precision or software accuracy, whereas deployment decisions in FastML depend on the synthesized datapath, compiler scheduling, and resource mapping. This study therefore presents an implementation-aware benchmark in which predictive performance and hardware efficiency are evaluated jointly under a common synthesis flow.

Submitted to 7th Fast Machine Learning for Science Conference (FastML 2026). Do not distribute.

## 2 Benchmark Design

### 2.1 BitNet formulation

BitNet constrains weights to binary or ternary values while restoring dynamic range through learned scaling factors. This replaces most dense multiplications along the quantized weight path with additions, subtractions, and zero operations, but introduces explicit scaling arithmetic. The benchmark therefore tests whether BitNet-style scaling improves the hardware-performance tradeoff relative to plain binary and ternary quantization after HLS synthesis.

### 2.2 Dataset and task

The OpenML `hls4ml_lhc_jets_hlf` benchmark [6] is used, consisting of approximately 830k balanced jets represented by 16 engineered observables across five classes: gluon (g), light quark (q), $W$, $Z$, and top (t).

The primary benchmark is defined as light-flavour QCD-like jets ($q/g$) versus jets labelled as $W$, $Z$, or top. This task serves as a compact public proxy for trigger-like hadronic object selection: the background corresponds to abundant QCD-like jets, while the signal combines multiple structured heavy-object classes. Compared with a single top-vs-QCD benchmark, the mixed $W/Z/t$ signal introduces greater intra-class variation and therefore provides a more stringent test of whether low-precision models retain discriminating power after FPGA-oriented quantization and synthesis.

High-level engineered features are deliberately used to isolate quantization and synthesis effects while avoiding confounding sequence or constituent-level inductive biases.

### 2.3 Models and synthesis

The primary benchmark uses a fixed 64–32–32 MLP topology. The benchmarked model families are dense MLPs, QKeras 7-bit fixed-point networks [1], HGQ mixed-precision networks [2], QKeras binary and ternary networks, BitNet binary networks, BitNet-1.58 sparse ternary networks, and XG-Boost BDT baselines [7]. HGQ denotes heterogeneous granularity quantization, where numerical precision can vary across model components. Tree-based baselines are included because they remain competitive for tabular trigger inference and provide strong latency-oriented FPGA references.

All neural models use identical features, preprocessing, fixed stratified 64/16/20% train/validation/test splits, random seeds (42–44), and synthesis constraints to isolate quantization and implementation effects. Neural MLP models are trained with binary cross-entropy using Adam. Full hyperparameters and exact training commands are provided in the released benchmark configuration files. Results are reported as mean $\pm$ standard deviation.

Neural models are converted and synthesized using hls4ml [8], while boosted decision trees are synthesized using Conifer [9]. All models are evaluated under a unified HLS C-synthesis flow targeting an AMD/Xilinx Virtex UltraScale+ VU13P FPGA with a 5 ns clock period and initiation interval II=1. Reported hardware metrics include latency, LUT, DSP, FF, and BRAM, with the primary benchmark focusing on latency, LUT, and DSP.

## 3 Results

### 3.1 Main benchmark

Table 1 summarizes predictive and hardware performance on the primary $q/g$ vs $W/Z/t$ benchmark. Signal efficiency near 1% background acceptance is included as a trigger-relevant low-background operating-point metric and is computed per seed from the ROC curve before averaging across seeds.

The main observations are: (i) BitNet consistently improves predictive performance over plain binary and ternary QKeras within this benchmark, but at increased LUT cost relative to plain ternary quantization; (ii) 7-bit QKeras retains dense-baseline performance with reduced latency, LUT, and DSP usage relative to the dense MLP; (iii) HGQ achieves the strongest neural hardware Pareto efficiency; and (iv) the tested unrolled BDT baseline provides the lowest synthesized latency.

Table 1: Main binary benchmark on OpenML 42468 for quark/gluon versus $W/Z/t$, using the fixed 64–32–32 MLP topology. The XGBoost baseline uses an ensemble of 100 trees with maximum depth 4. Values are reported as mean $\pm$ standard deviation over seeds 42–44. Signal efficiency is reported near 1% background acceptance (FPR $\approx 0.01$). Hardware metrics are VU13P, 5 ns, II=1 HLS C-synthesis estimates and do not include place-and-route. LUTs are reported in thousands.

| Model | Accuracy | AUC | $\epsilon_S$ @1% FPR | Latency | LUT [$10^3$] | DSP |
|---|---|---|---|---|---|---|
| Dense MLP | $0.8595 \pm 0.0003$ | $0.9325 \pm 0.0001$ | $0.5857 \pm 0.0021$ | $12.0 \pm 0.0$ | $183.5 \pm 0.3$ | 3455 |
| QKeras (7-bit) | $0.8595 \pm 0.0003$ | $0.9324 \pm 0.0001$ | $0.5813 \pm 0.0030$ | $9.0 \pm 0.0$ | $135.4 \pm 1.0$ | 665 |
| HGQ | $0.8538 \pm 0.0012$ | $0.9276 \pm 0.0002$ | $0.5711 \pm 0.0012$ | $6.3 \pm 0.6$ | $8.0 \pm 0.8$ | 0 |
| QKeras binary | $0.8294 \pm 0.0012$ | $0.8886 \pm 0.0063$ | $0.4319 \pm 0.1480$ | $20.0 \pm 2.0$ | $62.0 \pm 0.4$ | 0 |
| QKeras ternary | $0.8355 \pm 0.0023$ | $0.9025 \pm 0.0012$ | $0.4138 \pm 0.0127$ | $11.7 \pm 1.5$ | $28.3 \pm 3.2$ | 0 |
| BitNet binary | $0.8454 \pm 0.0003$ | $0.9176 \pm 0.0006$ | $0.4975 \pm 0.0103$ | $12.0 \pm 0.0$ | $88.3 \pm 0.3$ | 0 |
| BitNet-1.58 | $0.8489 \pm 0.0019$ | $0.9232 \pm 0.0013$ | $0.5345 \pm 0.0076$ | $9.0 \pm 0.0$ | $91.9 \pm 0.8$ | 0 |
| BDT (unrolled) | $0.8472 \pm 0.0003$ | $0.92075 \pm 0.00003$ | $0.5612 \pm 0.0006$ | $4.0 \pm 0.0$ | $74.0 \pm 0.1$ | 0 |

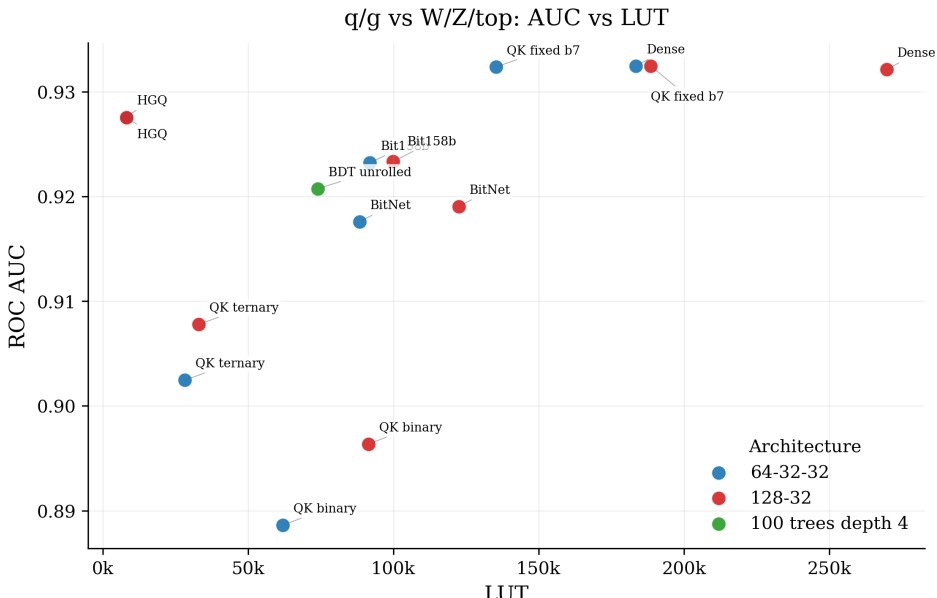

Figure 1: Pareto frontier of predictive performance (AUC) versus LUT usage for the binary $q/g$ versus $W/Z/t$ benchmark. HGQ occupies the strongest neural resource-efficiency region, achieving near-baseline performance at substantially reduced resource cost. BitNet-1.58 improves predictive performance over plain binary and ternary quantization at intermediate resource cost, while dense and moderate fixed-point models remain more resource-intensive.

## 3.2 Pareto analysis

To visualize the practical deployment tradeoffs, two Pareto frontiers are shown in Figures 1 and 2. AUC is used in the Pareto plots as a threshold-independent summary of predictive performance, while Table 1 additionally reports the trigger-relevant fixed-FPR operating point. Unlike Table 1, the Pareto plots include both the primary 64–32–32 topology and the secondary 128–32 robustness topology.

Figures 1 and 2 show predictive performance (AUC) versus LUT usage and synthesized latency, respectively.

Together, these Pareto frontiers highlight that different FastML methods occupy distinct optimal regions depending on whether predictive performance, latency, or resource efficiency is prioritized. In particular, HGQ provides the strongest neural hardware-efficiency tradeoff, while BitNet-1.58 offers the strongest ultra-low-bit compromise between predictive performance and implementation cost.

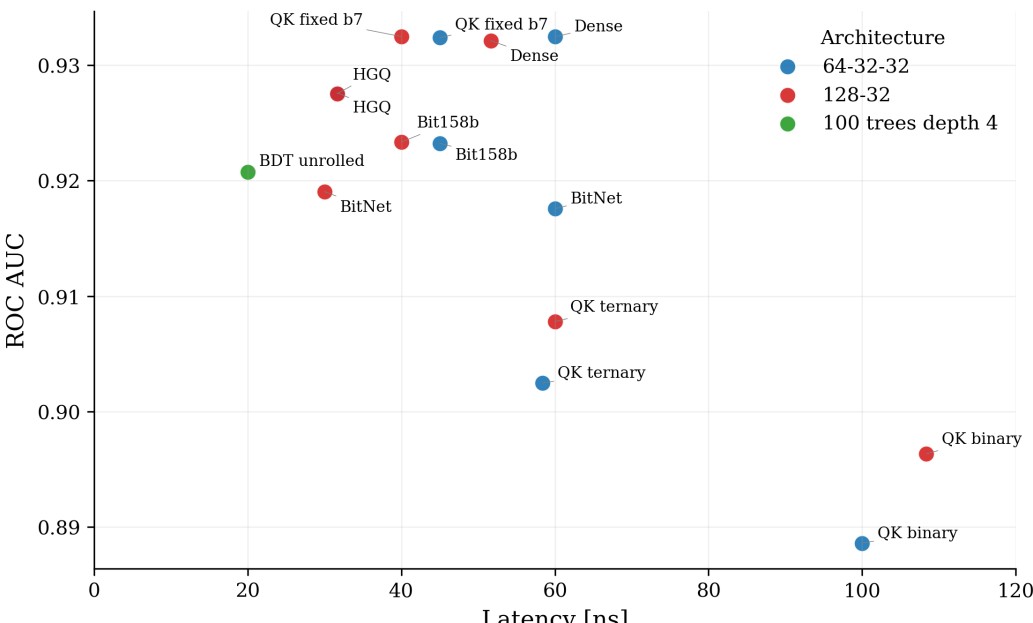

Figure 2: Pareto frontier of predictive performance (AUC) versus synthesized latency. Latency is reported in nanoseconds using the 5 ns HLS clock period, with the corresponding cycle counts reported in Table 1. The tested unrolled BDT baseline achieves the lowest latency, while HGQ and BitNet variants occupy competitive low-latency operating regions. The results show that lower arithmetic precision does not automatically guarantee lower synthesized latency.

## 3.3 Robustness

As a robustness check, the benchmark was repeated for a secondary $q/g$ versus top task and a wider 128–32 MLP topology. The same qualitative conclusions were observed: 7-bit QKeras retained dense-like predictive performance, HGQ remained the strongest neural resource-efficiency point, BitNet variants outperformed naive binary and ternary QKeras in AUC, and the tested unrolled BDT baseline remained the lowest-latency implementation. Full robustness tables are included in the released benchmark summaries.

## 4  Discussion and Conclusion

The results show that BitNet-style learned scaling provides clear predictive-performance improvements over naive ultra-low-bit quantization, particularly at binary and ternary precision. However, these gains come with implementation-dependent resource costs.

BitNet does not universally outperform advanced fixed-point approaches such as HGQ, which occupy stronger neural Pareto frontiers in both latency and resource efficiency. The central result is that theoretical operation-count reduction alone is insufficient to predict synthesized FPGA efficiency. Instead, implementation details such as scaling-factor realization, compiler scheduling, implementation pathway, and, for BitNet-1.58, structural sparsity can determine whether low-bit arithmetic gains survive synthesis.

For practical FastML deployment, the results favour fixed-point QAT for maximal performance retention, HGQ for neural resource efficiency, unrolled BDTs for minimal latency, and BitNet-style networks when ultra-low-bit inference is desired and scaling overhead can be implemented efficiently. DSP usage is reported separately because some implementations trade LUTs against DSPs.

**Limitations.** This benchmark is restricted to high-level engineered features and HLS C-synthesis estimates. Results may differ for constituent-level representations, trigger-native datasets, or after full place-and-route timing closure. HLS implementations also expose latency/resource tradeoffs, so the reported designs should be interpreted as synthesized reference implementations rather than globally optimal hardware realizations for each model family. The benchmark uses a single public high-level-feature dataset and three training seeds. Conclusions should therefore be interpreted as implementation-aware comparative evidence rather than a universal ranking across all trigger tasks. Final deployment decisions require post-synthesis or place-and-route validation.

## Code availability

Code, configurations, trained models, synthesis scripts, and benchmark summaries will be made publicly available at:

```
https://github.com/nairods/fastml_bitnet_benchmark
```

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

# NeurIPS Paper Checklist

